# Observation of Dirac Charge-Density Waves in Bi_2_Te_2_Se

**DOI:** 10.3390/nano13030476

**Published:** 2023-01-24

**Authors:** Adrian Ruckhofer, Giorgio Benedek, Martin Bremholm, Wolfgang E. Ernst, Anton Tamtögl

**Affiliations:** 1Institute of Experimental Physics, Graz University of Technology, Petersgasse 16, 8010 Graz, Austria; 2Dipartimento di Scienza dei Materiali, Università degli Studi di Milano-Bicocca, Via R. Cozzi 55, 20125 Milano, Italy; 3Donostia International Physics Center, University of the Basque Country, Paseo M. de Lardizabal 4, 20018 Donostia/San Sebastián, Spain; 4Centre for Materials Crystallography, Department of Chemistry and iNANO, Aarhus University, 8000 Aarhus, Denmark

**Keywords:** topological insulator, charge-density wave, electron-phonon coupling, spin-orbit coupling, atom-surface scattering

## Abstract

While parallel segments in the Fermi level contours, often found at the surfaces of topological insulators (TIs), would imply “strong” nesting conditions, the existence of charge-density waves (CDWs)—periodic modulations of the electron density—has not been verified up to now. Here, we report the observation of a CDW at the surface of the TI Bi${}_{2}$Te${}_{2}$Se(111), below ≈350K, by helium-atom scattering and, thus, experimental evidence for a CDW involving Dirac topological electrons. Deviations of the order parameter observed below 180K, and a low-temperature break of time reversal symmetry, suggest the onset of a spin-density wave with the same period as the CDW in the presence of a prominent electron-phonon interaction, originating from Rashba spin-orbit coupling.

## 1. Introduction

Charge-density waves (CDWs)—periodic modulations of the electron density—are a ubiquitous phenomenon in crystalline metals and are often observed in layered or low-dimensional materials [1,2,3,4,5,6]. CDWs are commonly described by a Peierls transition in a one-dimensional chain of atoms, which allows for an opening of an electronic gap at the nesting wavevector causing Fermi-surface nesting. However, it has been questioned whether the concept of nesting is essential for the understanding of CDW formation [1,7]. Instead, CDWs are often driven by strong electronic correlations and wavevector-dependent electron-phonon (e-ph) coupling [8]. Similarly, a nesting of sections of the Fermi surface can induce a periodic spin-density modulation, a spin-density wave (SDW) [9]. The possibility of a simultaneous appearance of both CDW and SDW order has been studied theoretically in earlier works [10] and an SDW was recently predicted for Weyl semi-metals [11].

The material class of topological insulators (TIs) has recently attracted extensive attention in a different context [12,13,14,15,16,17] due to their unique electronic surface states which involve a Dirac cone with spin-momentum locking [18,19]. Here, we report, for the first time, experimental evidence for a Dirac CDW on the surface of the TI Bi${}_{2}$Te${}_{2}$Se, i.e., a CDW involving Dirac topological electrons. Atom-scattering experiments reveal a CDW transition temperature $T_{CDW}=350\;K$ of the surface Dirac electrons, corresponding to a hexastar Fermi contour. The break of time-reversal symmetry of the CDW diffraction peaks observed at low temperature suggests a prominent role of Rashba spin-orbit coupling with the possible onset of an SDW below 180 K.

Archetypal TIs, such as the bismuth chalcogenides, share many similarities with common CDW materials, such as a layered structure (see Figure 1a) [20]. The hexagonal contours at the Fermi level often found in TIs also imply strong nesting which has led to speculations about the existence of CDWs in TIs [21]. Furthermore, the importance of charge order in the context of unconventional superconductivity in these systems has been subject to recent studies [22,23]. On the other hand, compared to other CDW materials, the topological surface states (TSS) of TIs, such as Bi${}_{2}$Te${}_{2}$Se, exhibit a characteristic spin polarisation, as studied together with the helical spin texture for the present material in [24,25]. Based on helium-atom scattering (HAS), it was recently shown that periodic charge-density modulations of the semimetal Sb(111) derive from multivalley charge-density waves (MV-CDW) due to surface pocket states [26]. While MV-CDWs are generally stabilised by electron-phonon (e-ph) interaction, different mechanisms can be responsible for CDW formation [1,7,8,27,28].

### Electronic Structure and Electron-Phonon Coupling

Among the bismuth chalcogenides, Bi${}_{2}$Te${}_{2}$Se is much less studied. The surface-dominated electronic transport [29,30,31,32,33], as well as the surface electronic band structure [24,34,35,36,37], have been subject to several investigations. Moreover, in terms of the electronic band structure it was shown that, for different Bi${}_{2-x}$Sb${}_{x}$Te${}_{3-y}$Se${}_{y}$ compositions, the Dirac point ($E_{D}$ in Figure 1c) moves up in energy with increasing *x* [34]. Tuning these stoichiometric properties and the doping of materials may give rise to nesting conditions between electron pocket or hole pocket states at the Fermi surface ($E_{F}$ in Figure 1c).

Figure 1c depicts the electronic surface band structure calculated by Nurmamat et al. [25] along the symmetry directions $\bar{\Gamma K}$ and $\bar{\Gamma M}$, revealing the TSS which form the Dirac cone. The Fermi level $E_{F}$ (horizontal purple line) in our present sample is located about 0.37eV above the Fermi energy, with respect to the calculations of Nurmamat et al. [25] and 0.43eV above the Dirac point, giving rise to the formation of quantum-well states at the Fermi surface. Moreover, a near-surface, two-dimensional electron gas (2DEG) with pronounced spin-orbit splitting can be induced on Bi${}_{2}$Te${}_{2}$Se by adsorption of rubidium [38]. Surface oxidation may occur at step-edge defects after cleaving [39], but Bi${}_{2}$Te${}_{2}$Se seems to be less prone to the formation of a 2DEG from rest-gas adsorption compared to other TIs [40], as shown in angle-resolved photo-emission measurements of the present samples [31]. Dirac fermion dynamics in Bi${}_{2}$Te${}_{2}$Se were subject to a recent study by Papalazarou et al. [41].

One reason for Bi${}_{2}$Te${}_{2}$Se being less studied than the binary bismuth chalcogenides might be the difficulty in synthesising high-quality single crystals, which originates from the internal features of the specific solid-state composition and phase separation in Bi${}_{2}$Te${}_{2}$Se [42]. In this work, we present a helium-atom scattering (HAS) study of Bi${}_{2}$Te${}_{2}$Se, which is actually phase II of Bi${}_{2}$Te${}_{3-x}$Se${}_{x}$(111) with x=1 according to [42], as derived from the surface lattice constant a=4.31Å, measured by HAS for the structure shown in Figure 1. Since helium atoms are scattered off the surface electronic charge distribution, HAS [43,44] provides access to the surface electron density [45,46] and is, therefore, a perfect probe for experimental studies of TIs since the TSS properties are often mixed up with those of bulk-states [47,48,49]. As the surface electronic transport properties of TIs at finite temperature, as well as the appearance of CDWs, are influenced by the interaction of electrons with phonons, the e-ph coupling described in terms of the mass-enhancement factor λ has been subject to several studies [47,50,51,52,53,54,55,56]. For Bi${}_{2}$Te${}_{2}$Se, it was reported that the electron-disorder interaction dominates scattering processes with λ=0.12 [57], in good agreement with the value found from HAS [55].

## 2. Experimental Details

The experimental data for this work was obtained using a HAS apparatus, where a nearly monochromatic beam of ${}^{4}$He is scattered off the sample surface in a fixed source-sample-detector geometry (for further experimental details, see [58] and Appendix A). The scattered intensity of a He beam in the range of 10–15 meV is then recorded as a function of the incident angle $\vartheta_{i}$ with respect to the surface normal, which can be modified by rotating the sample in the scattering chamber. The momentum transfer parallel to the surface ∆K, upon elastic scattering, is given by
(1)$$∆K=\left|k_{i}\right|\left(\sin\vartheta_{f}-\sin\vartheta_{i}\right),$$
with $k_{i}$, the incident wavevector, and $\vartheta_{i}$ and $\vartheta_{f}$ the incident and final angle, respectively.

The Bi${}_{2}$Te${}_{2}$Se sample was grown by the Bridgman–Stockbarger method, as detailed in [42], and further characterised using powder X-ray diffraction (PXRD), Seebeck microprobe measurements and inductively coupled plasma atomic emission spectroscopy. While the chemical composition varies along the grown crystal rod, with the Se content *y* in the formula Bi${}_{2}$Te${}_{x}$Se${}_{y}$ changing along the distance from the growth starting point, as illustrated in [42], experimental PXRD shows that smaller sections of the entire crystal boule correspond to a single phase. The relation between the Se content *y* and the cell parameters *a* and *c*, allows to assign the here used section of the crystal rod to phase II of Bi${}_{2}$Te${}_{3-x}$Se${}_{x}$(111) with x≈1, according to the above-mentioned surface lattice constant a=4.31Å from HAS diffraction scans.

As also outlined in [42], the composition along individual sections of the crystal rod is uniform, with the cell parameters *a* and *c* changing abruptly at specific points along the rod. The single-crystallinity of the present sample is further supported by HAS and low-energy electron diffraction measurements in the present work. Finally, the Bi${}_{2}$Te${}_{2}$Se sample was cleaved in situ, in a load-lock chamber [59] prior to the experiments. Due to the weak bonding between the quintuple layers in Figure 1a, the latter gives access to the (111) cleavage plane with a Te termination at the surface (Figure 1b, see also Appendix A).

## 3. Results and Discussion

### 3.1. Surface CDW Order

Figure 2 shows the scattered He intensity versus momentum transfer ∆K (Equation 1) with the scans taken at different incident-beam energies $E_{i}$ along the $\bar{\Gamma M}$ (a) and $\bar{\Gamma K}$ (b) directions. Along $\bar{\Gamma M}$, there appear sharp additional peaks (illustrated by the vertical dashed lines) next to both the specular and the first-order diffraction peaks (vertical dash-dotted lines) at an average spacing of about $0.18\;{Å}^{-1}$ with respect to the diffraction peaks (Figure 2 and Figure 3b at an enlarged scale). The fact that these satellite peaks appear at the same momentum transfer $∆K\approx \pm 0.18\;{Å}^{-1}$ with respect to the specular, as well as to the first-order diffraction peaks, independently of the incident energy, shows that they are neither caused by bound-state resonances [45] nor other artifacts (see the section on CDW satellite peaks in Appendix B, as well as additional scans in Figure A1), but have, necessarily, to be ascribed to a long-period surface superstructure of the electron density, i.e., a surface CDW.

These observations are consistent with the theoretical surface-band structure calculated by Nurmamat et al. [25] (Figure 1c) and, more specifically, with the distribution in parallel wavevector space of the spin polarisation perpendicular to the surface for the states at the Fermi level, when re-scaled for a Fermi energy $E_{F}=0.43\;$meV above the Dirac point. As appears in Figure 3a, the re-scaled spin distribution (red and blue for spin-up and spin-down, respectively) exhibits extended parallel segments of equal spin separated by the nesting wavevector $g_{\mathrm{CDW}}=0.18\;{Å}^{-1}$ along the $\bar{\Gamma M}$ directions. No such nesting occurs along the $\bar{\Gamma K}$ directions, in agreement with experiments.

The position of the Fermi level which accounts for the present data falls near the bottom of the band of surface quantum-well states (grey circular region in Figure 3a), allowing for a continuum of small-wavevector phonon-induced transitions across the Fermi level, which may possibly be associated with the observed diffraction structure around the HAS specular peak.

Figure 3 shows that the hexastar shape provides a spin-allowed nesting which corresponds to the observed $g_{\mathrm{CDW}}$ periodicity. In contrast to the hexastar, for a hexagonal shape, as found in several TIs or as also observed on Bi(111), opposite sides of the Fermi contour exhibit opposite spins—a situation which forbids the pairing needed for a CDW formation, but leaves the possibility of an SDW [60,61]. Finally, the transmission of momentum and energy to the lattice for the spin-allowed transition across the hexastar occurs, via the excitation of virtual electron-hole pairs, if one assumes an Esbjerg–Nørskov form of the atom-surface potential based on a conducting surface.

### 3.2. CDW Temperature Dependence

In the following, we consider the temperature dependence of the CDW diffraction peaks and the CDW critical temperature $T_{CDW}$. Upon measuring the scattered intensities as a function of surface temperature, it turns out that the intensity of the satellite peaks decreases much faster than the intensity of the specular peak. As shown in Appendix B (Figure A2), when plotting both peaks in a Debye–Waller plot, the slope of the satellite peak is clearly steeper than the one for the specular peak.

Based on the theory of classical CDW systems, the square root of the integrated peak intensity can be viewed as the order parameter of a CDW [62,63]. Figure 4b shows the temperature dependence of the square root of the integrated intensity for the $-g_{\mathrm{CDW}}$ peak on the left-hand-side of the specular peak (see right panel of Figure 4b for several scans). In order to access the intensity change relevant to the CDW system, as opposed to the intensity changes due to the Debye–Waller factor [64], the integrated intensity I(T) has been normalised to that of the specular beam [65]—a correction which is necessary in view of the low surface Debye temperature of Bi${}_{2}$Te${}_{2}$Se(111) [55]. The temperature dependence of the order parameter $\sqrt{I(T)}$ can be used to determine the CDW transition temperature $T_{CDW}$ and the critical exponent β belonging to the phase transition by fitting the power law
(2)$$\sqrt{\frac{I(T)}{I(0)}}={\left(1-\frac{T}{T_{CDW}}\right)}^{\beta }\;,$$
to the data points in Figure 4b. Here, I(0) is the extrapolated intensity at 0K. The fit is represented by the green dashed line in Figure 4b, resulting in $T_{CDW}=\left(350\;\pm 10\right)$ K and β=(0.34±0.02).

The exact peak position and width of the satellite peak was determined by fitting a single Gaussian to the experimental data. The right panel of Figure 4a shows a shift in the satellite peak position to the right with increasing surface temperature, i.e., $g_{\mathrm{CDW}}$ decreases with increasing temperature, as illustrated by the grey line. Such a temperature dependence confirms the connection of the satellite peaks with the surface electronic structure. A shift of the Dirac point to lower binding energies with increasing temperature and, thus, a decrease in $k_{F}$ has been observed both for Bi${}_{2}$Te${}_{2}$Se(111) [67] and Bi${}_{2}$Se${}_{3}$(111) [68]. As reported by Nayak et al. [67], the temperature-dependent changes in the electronic structure at $E_{F}$ occur due to the shift of the chemical potential in the case of *n*-type Bi${}_{2}$Te${}_{2}$Se(111). Moreover, a strong temperature dependence of the chemical potential has also been observed for other CDW systems [69,70] and semiconductors [71]. It is noted, however, that, in the present case, changes in $g_{\mathrm{CDW}}$ with temperature, as well as in the peak area, are concentrated in a region around 170K, which, by itself, suggests another phase transition.

### 3.3. Diffraction and the Role of Spin-Orbit Coupling

The surprising disappearance of the $+g_{\mathrm{CDW}}$ diffraction peak observed at low temperature (118 K) can indeed be related to the apparent phase transition occurring at about 170–180 K (illustrated by the orange arrow in Figure 4), possibly a spin ordering within the CDW, i.e., an SDW with the same period. The latter is indicated by a rapid shift in the $-g_{\mathrm{CDW}}$ diffraction peak position around 170 K, corresponding to a slight contraction of the CDW period (right panel of Figure 4a) as a possible effect of spin-ordering. The CDW order parameter, expressed by ${\left(1-T/T_{CDW}\right)}^{\beta }$ actually shows a small deviation from this law below about 180K ($T_{SDW}$ in Figure 4b). As explained above, the He-atom diffraction process from the CDW occurs via parallel momentum transfer to the surface electron gas via an electron-hole excitation between a filled and an empty state of equal spin and well nested at the Fermi level. Thus, the unidirectionality of the process at low temperature suggests a prominent role of the Rashba term in the presence of spin ordering with strong implications for the e-ph contribution.

The latter is in line with Guan et al. [72], who reported a large enhancement of the e-ph coupling in the Rashba-split state of the Bi/Ag(111) surface alloy. The larger overlap of He atoms with CDW maxima also selects electrons with the same spin, because the SDW and CDW exhibit the same period. Considering, in the present case, a free-electron Hamiltonian [73],
(3)$$-E_{D}+\frac{p^{2}}{2m^{*}}+\alpha_{R}\;\sigma \;\cdot \;\left(p\times \hat{z}\right)\;,$$
where $-E_{D}$ is the energy of the Dirac point below the Fermi energy $E_{F}=0$, p is the surface electron momentum and $m^{*}$ its effective mass, σ is the spin operator, $\hat{z}$ is the unit vector normal to the surface and $\alpha_{R}$ is the Rashba constant. The modulation of the Rashba term
(4)$$\frac{\partial \alpha_{R}}{\partial A_{qs}}\;A_{qs}\;\sigma \;\cdot \;\left(q\times \hat{z}\right)\;,$$
produced by a phonon of momentum q, branch index *s* and normal mode coordinate $A_{qs}$ is viewed as the main source of e-ph interaction [74], causing inter-pocket coupling $\left(∆p=q=\pm g_{\mathrm{CDW}}\right)$ and CDW gap opening.

In a diffraction process, the exchange of parallel momentum between the scattered atom and the solid centre-of-mass is mediated by a virtual electron inter-pocket transition $|k,n\rangle \to |k+q,n^{\prime }\rangle$ weighed by the difference in Fermi–Dirac occupation numbers $f_{q,n}-f_{k+q,n^{\prime }}$. While the process $q=-g_{\mathrm{CDW}}$ virtually casts the electron from a pocket ground state at the Fermi level to an empty excited state across that gap, the process $q=+g_{\mathrm{CDW}}$ would virtually send the electron, because of the Rashba term, to a lower energy state and is, therefore, forbidden at low temperature.

Such a scenario is equivalent to saying that the SDW–CDW entanglement makes HAS sensitive to the spin orientation via its temperature dependence. The inter-pocket electron transition across the gap accompanying a CDW diffraction of He atoms via the modulation of the Rashba term may only occur in one direction. Since the gap energy is of the order of room temperature, and at this temperature the spin ordering is removed, the above selection rule is relaxed and the diffraction peaks are observed in both directions (on both sides of the specular peak in Figure 3b).

We note that the nesting condition and the changes between different Fermi level contours (e.g., hexagonal vs. hexastar shapes) depend strongly on the position of the Fermi level [40,75,76] and may, thus, be highly sensitive to the doping situation of the specific sample [77]. While the size of the CDW gap, as inferred from the slight asymmetry between $+g_{\mathrm{CDW}}$ and $-g_{\mathrm{CDW}}$, does not seem to be large enough to be resolved in ARPES [21,70,77], HAS satellite diffraction peaks clearly indicate an additional long-period component of the surface charge-density corrugation.

## 4. Summary and Conclusions

In summary, we have provided evidence by means of helium-atom scattering, of a surface charge-density wave in Bi${}_{2}$Te${}_{2}$Se(111) occurring below 345 K and involving Dirac topological electrons. The CDW diffraction pattern is found to reflect a spin-allowed nesting across the hexastar contour at the appropriate Fermi level, re-scaled from previously reported ab initio calculations [25]. The CDW order parameter has been measured down to 108K and found to have a critical exponent of 1/3. The observation of a time-reversal symmetry break at low temperature, together with deviations from the critical behaviour below about 180K, are interpreted as being due to the onset of a spin-density wave with the same period as the CDW in the presence of a prominent electron–phonon interaction originating from the Rashba spin-orbit coupling.

While it is difficult to make definitive statements about the generality of our observations, we anticipate that, by tuning the stoichiometric properties and doping level of topological insulators, thus changing the position of the Dirac point and possible nesting conditions, the condition for the CDW order may be changed or shifted to a different periodicity. It is, thus, expected that, from further experiments and validation, one may be able to evolve phase diagrams for Dirac CDWs as a function of stoichiometry, doping and Fermi-level position. Taken together the results promise also to shed light on previous experimental and theoretical investigations of related systems and how the CDW order affects lattice dynamics and stability. These also include possible connections between the CDW order and superconductivity [78,79,80], as well as the influence of certain energy dissipation channels on molecular transport [81,82].

## Figures and Tables

**Figure 1 nanomaterials-13-00476-f001:**
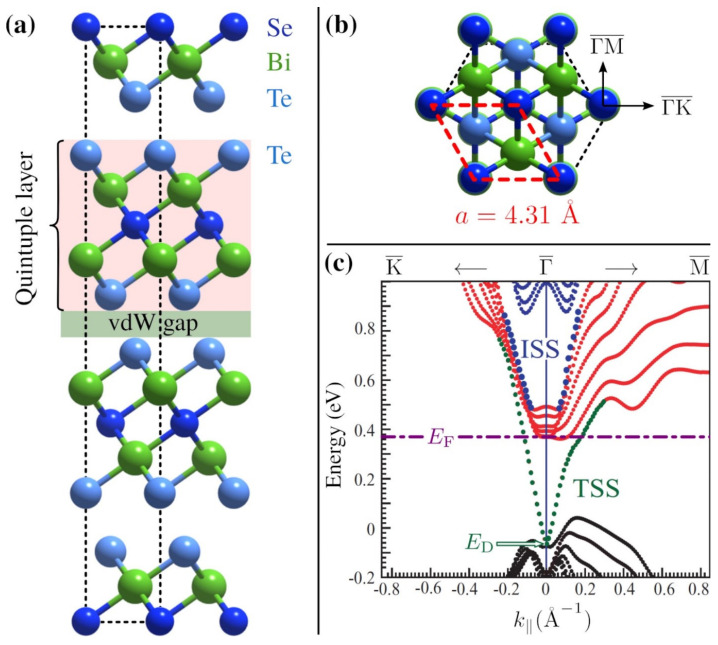
(**a**) Side view of the conventional hexagonal unit cell of Bi${}_{2}$Te${}_{2}$Se. The unit cell consists of three quintuple layers, each of which is terminated by a Te layer. (**b**) The (111) cleavage plane, according to rhombohedral notation, with the red rhombus highlighting the hexagonal surface unit cell of the (0001) plane in hexagonal notation. (**c**) Surface band structure of Bi${}_{2}$Te${}_{2}$Se(111) calculated by Nurmamat et al. [25] (reproduced with permission, copyright 2013 by the American Physical Society) along the symmetry directions $\bar{\Gamma K}$ and $\bar{\Gamma M}$. TSS labels the topological surface states forming the Dirac cone, while internal (quantum-well) surface states (ISS) are also found in the gap above the conduction band minimum. In the present sample, the Fermi level $E_{F}$ (purple dash-dotted line) is located about 0.43eV above the Dirac point $E_{D}$ and 0.37eV above the Fermi energy, i.e., the zero ordinate in [25].

**Figure 2 nanomaterials-13-00476-f002:**
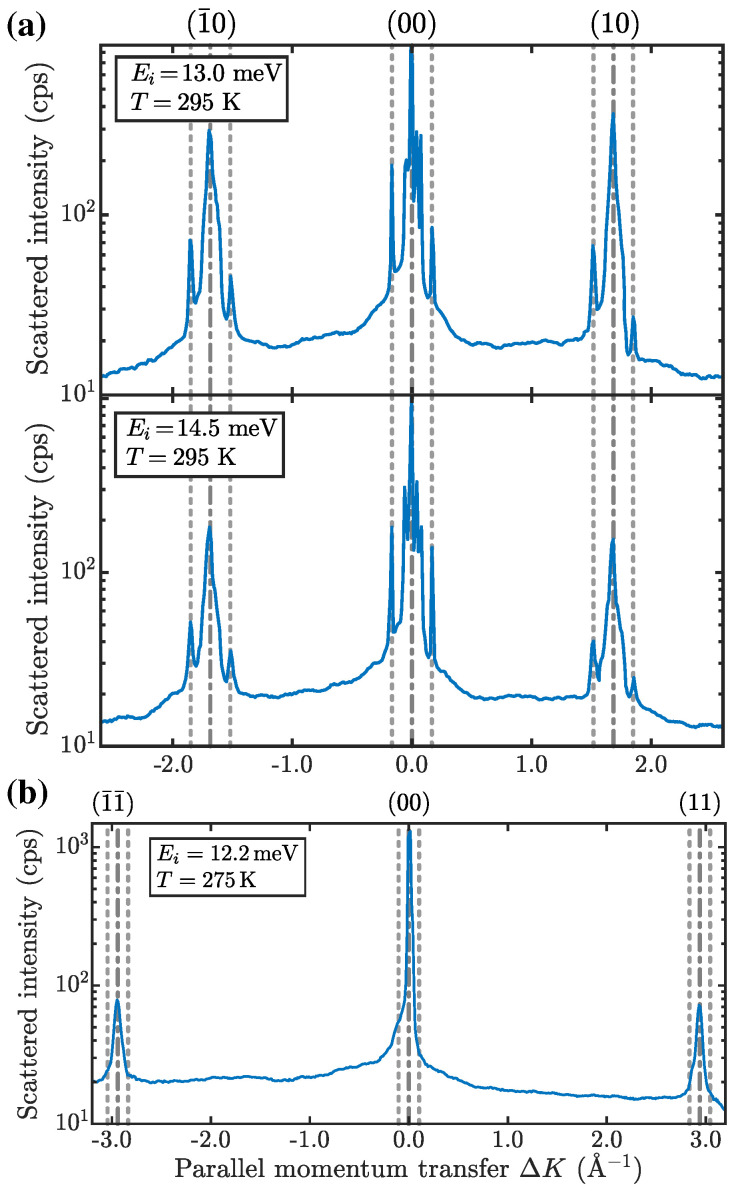
The CDW periodicity becomes evident in diffraction scans (logarithmic scale) of Bi${}_{2}$Te${}_{2}$Se(111). (**a**) Scans along $\bar{\Gamma M}$ taken at room temperature show satellite peaks (illustrated by the grey dashed vertical lines) next to the specular and first-order diffraction peaks (grey dash-dotted vertical lines). (**b**) Same for the $\bar{\Gamma K}$ direction; in this direction no evident satellite peaks are observed close to the diffraction peaks.

**Figure 3 nanomaterials-13-00476-f003:**
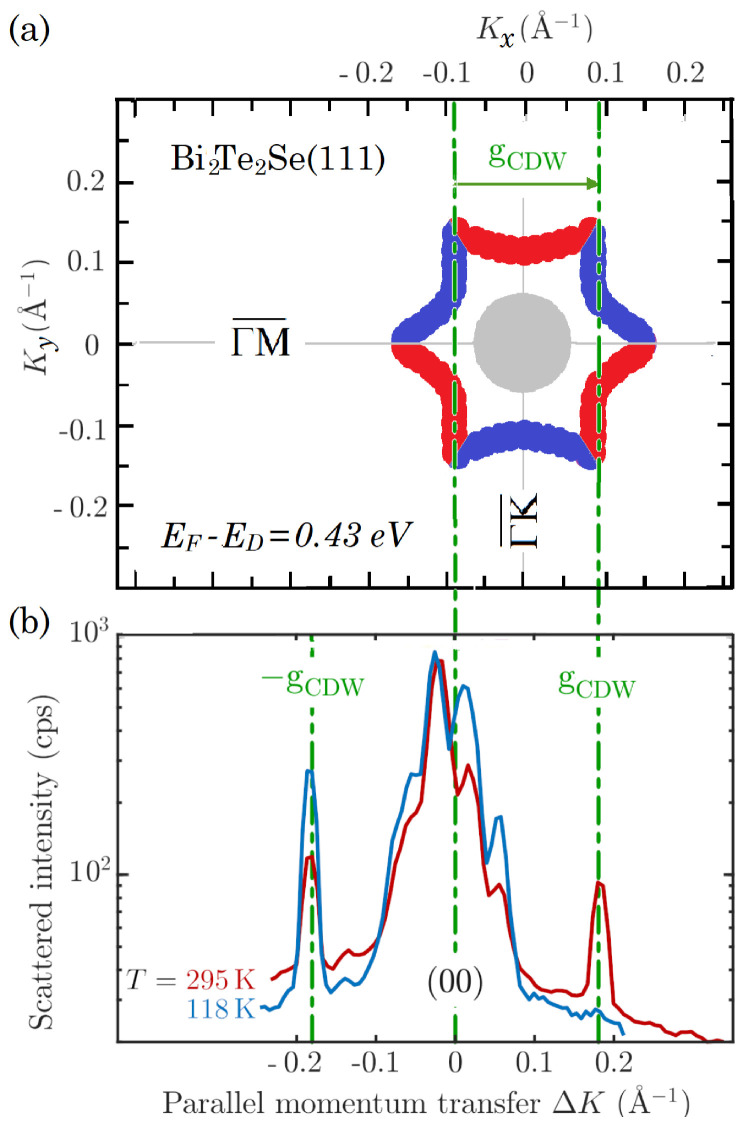
Illustration of the spin-allowed nesting condition in (**a**), giving rise to the sharp CDW satellite peaks at $\pm g_{\mathrm{CDW}}$ around the (00)-reflection in (**b**). HAS diffraction scans of Bi${}_{2}$Te${}_{2}$Se(111) along the $\bar{\Gamma M}$ direction at room (295 K) and low (118 K) temperature in (**b**), show two sharp peaks at wavevectors $\pm g_{\mathrm{CDW}}$, indicative of a surface CDW. While, at room temperature, the two diffraction peaks have about the same intensity, at 118K the peak intensity at $-g_{\mathrm{CDW}}$ is almost doubled at the expense of the peak at $+g_{\mathrm{CDW}}$, which is vanishing. (**a**) The contours of the Dirac surface states at the Fermi level, derived from Nurmamat et al. surface band structure calculations [25] for a Fermi level 0.43eV above the Dirac point (see Figure 1c), form a hexastar (red and blue branches for states with spin-up and spin-down normal to the surface, respectively), with a clear nesting wavevector $g_{\mathrm{CDW}}$ connecting parallel contours of equal spin. No such nesting occurs in the $\bar{\Gamma K}$ direction.

**Figure 4 nanomaterials-13-00476-f004:**
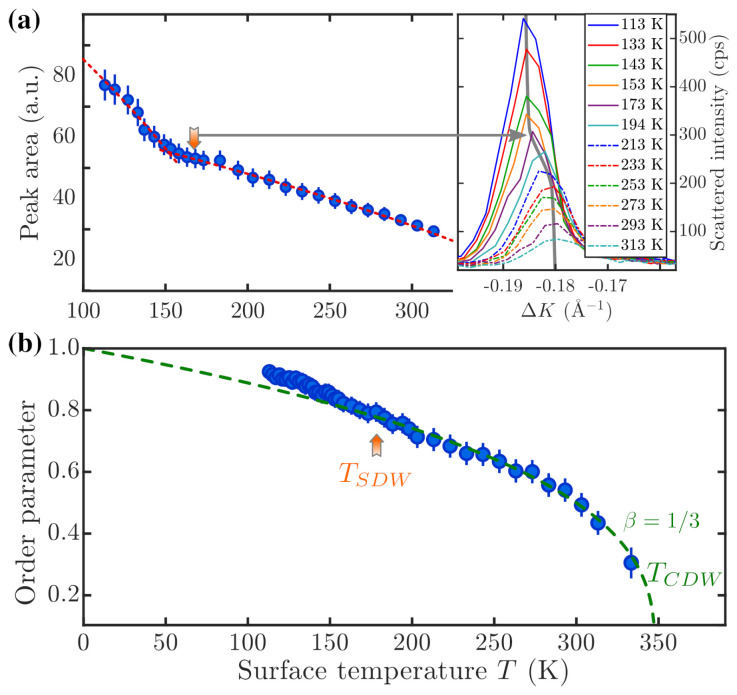
Peak intensity and order parameter of the surface CDW (blue circles), as extracted from the temperature dependence of the $-g_{\mathrm{CDW}}$ CDW peak in Figure 3b (see text). The right panel in (**a**) shows several scans, illustrating a rapid shift in the satellite peak, occurring at around 180K, from a momentum transfer of $0.18\;{Å}^{-1}$ to $0.186\;{Å}^{-1}$. Together with the corresponding deviation of the order-parameter fit below about 180K ($T_{SDW}$) in (**b**) and the time-reversal symmetry break (missing $+g_{\mathrm{CDW}}$ peak at low *T* in Figure 3b), it suggests the onset of a spin ordering (a spin-density wave), which allows, through the Rashba effect, for a parallel momentum transfer only in one direction. (**b**) The fit of the order parameter with the fluctuation critical exponent β=1/3 [66] (green dashed line) yields a critical temperature $T_{CDW}$ of about 350K. The red dashed line in (**a**) illustrates separate fits to the peak intensity in the two temperature regimes with two critical temperatures $T_{SDW}$ and $T_{CDW}$, respectively.

## Data Availability

Experimental data supporting the results are available from the corresponding author upon reasonable request.

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
