# Peer review of "Observation of Dirac Charge-Density Waves in Bi2Te2Se"

_nanomaterials, 2023, doi:10.3390/nano13030476_

Round 1
Reviewer 1 Report
The authors reported herein the surface charge density wave in Bi2Te2Se(111) appeared below 345 K involving Dirac electrons as revealed by helium atom scattering measurements. They also observed a time reversal breaking at low temperature together with deviations from the critical behavior below ~ 180 K, which was ascribed to a spin density wave behavior with the same periodicity as the charge density wave in presence of large electron-phonon coupling due to Rashba spin-orbit coupling. The interesting results definitely will excite interest in the research field. The measurements and characterizations are solid and full support the conclusions. The analysis is reasonable. The paper was well written and organized. I therefore would like to recommend a publication of this paper. My only concern is that the authors are better to show some basic chracterizations results of their used samples, for sample, the phase purity and compositions, etc., because high-quality Bi2Te2Se crystal is difficult to be synthesized and the quality actually plays important role in influencing the physical properties.
Reviewer 2 Report
In the manuscript, the authors carried out a helium atom scattering study of Bi2Te2Se. The additional peaks along indicate the presence of surface charge (CDW) with wavevector = 0.18 Å-1 below ~ 350 K, which is consistent with the theoretical calculations. At low temperature, the deviations from critical behaviour of the order parameter and the vanish of - peak are considered to be come from the occurrence of spin density wave (SDW). This study and the analysis of results is clear. I recommend accepting this manuscript and publishing on Nanomaterials. Before accepting this manuscript, however, there are some comments on this work need to be addressed.
1. How did the author define the (111) plane, which should be (00l) plane in my opinion?
2. The study lacks basic characterization of crystals. How did the authors determine that the crystals of phase II of Bi2Te3-xSex? How did the authors determine the Fermi energy of 0.43 meV above the Dirac point, which is inconsistent with the ARPES results of ~ 0.37 meV in ref[42]?
3. In Figure 4(a), the meaning of the red dotted line is not marked. The temperature dependence of CDW peaks which labeled as Figure 3(c) is missing. In particular, when did the peak vanish as the temperature decreases?
4. There are some typos. For example, it should be “electron/ hole pocket” on line 52-53.

Reviewer 3 Report
The paper reports on experimental evidence for surface charge density wave involving Dirac topological electrons by means of helium atom scattering. The paper could be of interest to readers of the journal. However, information on the sample used in the study is insufficient. I suggest major revision before ready for publication.
1. The information on the sample should be added in the section “2. Experimental Details”. In the section “A1. Experimental Details”, it is only described that “Details about the sample growth procedure can be found in Mi et al. [42]”. However, in Ref.42, it was revealed that the chemical composition and concentration of point defects vary along the Bi2Te2Se crystal rod grown by the Stockbarger method.
2. In page2, lines56-58, it is described “The Fermi level EF (horizontal purple line) in our present sample is located about 0.37 eV above the Fermi level with respect to calculations of Nurmamat et al.[25] and 0.43 eV above the Dirac point…”. The results of ARPES should also be added, in order to confirm the validity of the Fermi level and also the shape of contours of the surface states at the Fermi level shown in Figure 3(a).
3. In page6, caption of Figure 4, “Figure 3(c)” should be corrected to “Figure 3(b)”.
4. In page7, “4. Summary and Conclusion” section, abbreviations, HAS, CDW, and SDW, should be used instead of “helium atom scattering”, “charge density wave”, and “spin density wave”, respectively.
I hope these comments will be helpful.
Round 2
Reviewer 3 Report
The authors have addressed most of the reviewer's comments and the manuscript can be accepted for publication.